# Role of FABP5 in T Cell Lipid Metabolism and Function in the Tumor Microenvironment

**DOI:** 10.3390/cancers15030657

**Published:** 2023-01-20

**Authors:** Rong Jin, Jiaqing Hao, Jianyu Yu, Pingzhang Wang, Edward R. Sauter, Bing Li

**Affiliations:** 1Department of Microbiology and Immunology, University of Louisville, Louisville, KY 40202, USA; 2NHC Key Laboratory of Medical Immunology, Department of Immunology, School of Basic Medical Sciences, Peking University, Beijing 100191, China; 3Department of Pathology, University of Iowa, Iowa City, IA 52242, USA; 4Division of Cancer Prevention, National Institutes of Health/National Cancer Institute, Bethesda, MD 20892, USA

**Keywords:** FABP5, lipid metabolism, T cells, tumor microenvironment, immunotherapy, obesity

## Abstract

**Simple Summary:**

T cells infiltrating in the tumor microenvironment play a critical role in anti-tumor immunity. A well-balanced metabolism in T cells determine their function and fate. In this review, we summarize an emerging role of fatty acid binding protein 5 (FABP5, also known as epidermal FABP, or E-FABP), a cytosolic lipid chaperone, in facilitating fatty acid uptake, transport, and metabolism and regulating the differentiation and function of different T cell subsets. Therefore, FABP5 represents a new lipid sensor in determine T cell lipid metabolism and function in the tumor microenvironment.

**Abstract:**

To evade immune surveillance, tumors develop a hostile microenvironment that inhibits anti-tumor immunity. Recent immunotherapy breakthroughs that target the reinvigoration of tumor-infiltrating T lymphocytes (TIL) have led to unprecedented success in treating some cancers that are resistant to conventional therapy, suggesting that T cells play a pivotal role in anti-tumor immunity. In the hostile tumor microenvironment (TME), activated T cells are known to mainly rely on aerobic glycolysis to facilitate their proliferation and anti-tumor function. However, TILs usually exhibit an exhausted phenotype and impaired anti-tumor activity due to the limited availability of key nutrients (e.g., glucose) in the TME. Given that different T cell subsets have unique metabolic pathways which determine their effector function, this review introduces our current understanding of T cell development, activation signals and metabolic pathways. Moreover, emerging evidence suggests that fatty acid binding protein 5 (FABP5) expression in T cells regulates T cell lipid metabolism and function. We highlight how FABP5 regulates fatty acid uptake and oxidation, thus shaping the survival and function of different T cell subsets in the TME.

## 1. Introduction

Tumor immunotherapy, particularly T lymphocyte-based immunotherapy, has achieved clinical benefits in multiple cancers, including metastatic melanoma [1], lung [2], leukemia [3], bladder [4], ovarian [5] and sarcoma [6]. There are primarily three strategies that modulate T cell activity for tumor therapies: (1) enhancing T cell/tumor recognition by genetic modification of T cell receptors (TCRs), such as transducing non-tumor specific T cells with tumor-targeting chimeric antigen receptors (CARs) to transform them into tumor-specific killers [7]; (2) adoptive cell transfer (ACT) of ex vivo expansion of tumor-infiltrating lymphocytes to promote tumor regression [8]; and (3) reinvigorating the tumor-killing activity of tumor-specific T cells in the tumor microenvironment (TME). Utilizing the immune checkpoint blockade (ICB) that targets programmed cell death protein 1 (PD-1)/its receptor (PD-L1), or cytotoxic T cell associated protein 4 (CTLA4), unleashes the anti-tumor effect of T cells in the TME, leading to therapeutic benefits in treating some tumors resistant to conventional therapy [9,10]. However, only a small percentage of cancer patients with solid tumors respond to T cell-based immunotherapy. Multiple extrinsic and intrinsic factors, including the presence of different T cell subsets, T cell activation signals, and the metabolic status of T cells in the TME, seem to be critical in determining a cancer patient’s response to ICB [11,12,13]. In this review, we introduce our current understanding of T cell development and activation and discuss how T cell-activating signals regulate T cell metabolic pathways, which in turn govern T cell differentiation and survival. Further, we summarize the emerging role of epidermal fatty acid binding protein (E-FABP, or FABP5), a previously underappreciated lipid sensor in T cells, in modulating fatty acid metabolism and responses in different T cell subsets [14,15,16], which highlights a new concept of how the modulation of lipid metabolism by FABP5 influences the anti-tumor function of T cells in the TME.

## 2. T Cell Development in the Thymus and Activation in the Periphery

Bone marrow-derived T-cell precursors travel to the thymus, where they undergo three sequential developmental stages based on the cell-surface expression of CD4 and CD8 markers: CD4^−^CD8^−^ double-negative (DN), CD4^+^CD8^+^ double-positive (DP) and CD4^+^ or CD8^+^ single-positive (SP). DN and DP thymocytes develop in the thymic cortex, whereas SP thymocytes mature in the thymic medulla. After undergoing a series of complicated phenotypic/functional maturation and positive and negative selections (Figure 1), DN cells develop into SP CD4^+^ or CD8^+^ T cells and egress to the peripheral T-cell repertoire as naïve CD4^+^ or CD8^+^ T cells [17,18].

Naïve CD4^+^ or CD8^+^ T cells circulate in the blood, the peripheral lymphoid organs and different tissues [19,20]. Upon encountering antigens presented by major histocompatibility complex (MHC) molecules on the surface of antigen presenting cells (APCs), T cells recognize specific antigens through T cell receptors (TCR) by forming a TCR/antigen/MHC complex. CD4 or CD8 molecules on naïve T cells strengthen the stability of the complex by binding to MHCII or MHCI on APCs. The specific recognition of TCR/antigens initiates the first signal for T cell activation. However, full T cell activation requires additional co-stimulatory signals. For example, constitutive expression of CD28 on T cells binds to CD80 or CD86 on APCs, providing secondary signals to trigger T cell proliferation. Antigen-encountered APCs can further activate T cells by inducing expression of new co-stimulatory receptors (e.g., CD137, ICOS) and additional signals. Besides signals from TCR/antigen recognition and co-stimulatory ligands, other factors, such as environmental cytokine signals, also affect T cell activation and differentiation. It is well known that naïve CD4^+^ T cells can differentiate into different helper effector subsets, including Th1, Th2, Th17, regulatory T cells (Treg) and follicular T cells (Tfh) [21], depending on the cytokine milieu to which they are exposed. Naïve CD8^+^ T cells can differentiate into effector cytotoxic T lymphocytes (CTLs), performing anti-infection and anti-tumor responses [20,22]. After antigen clearance, the majority of terminally differentiated effector T cells undergo apoptosis and die, and the remaining cells persist as long-lived memory T cells (Tm), which confer rapid recall responses if encountering the same antigens again. Notably, during the life cycle of T cells, from development in the thymus to patrolling throughout the body, from quiescent naïve state to activated and/or memory status, external signals induce appropriate T cell activation by coordinating different metabolic pathways inside T cells.

## 3. External Signals Direct Internal Metabolic Pathways during T Cell Activation

There is a growing realization that external signals trigger T cell activation and function by coordinating internal metabolic pathways [23,24,25]. T cells mainly utilize two key metabolic pathways, aerobic glycolysis and mitochondrial oxidative phosphorylation (OXPHOS), to fulfill their metabolic needs during their activation processes [26,27]. After leaving the thymus, CD4^+^ and CD8^+^ naïve T cells remains quiescent in the periphery [28]. Homeostatic cytokines, such as IL-7, provide external survival signals through activation of the AKT/glucose transporters’ (e.g., Glut1) pathway in naïve T cells [29]. Glucose uptake in naïve T cells is converted into pyruvate in the cytoplasm and fully oxidized in the mitochondria through the OXPHOS pathway to maintain their metabolic quiescence [30]. Upon encountering antigen signal stimulation (e.g., TCR/CD28), naïve T cells require a dramatic increase in nutrient uptake to support the highly energetic needs necessary for T cell proliferation and differentiation [31]. It has been shown that TCR/CD28 signals regulate multiple metabolic pathways. For instance, TCR/CD28 signals not only induce the upregulation of transporters for glucose (e.g., Glut1) and amino acids (e.g., ASCT2 for glutamine and Slc7a5 and CD98 for sodium-coupled neutral amino acids) to enhance exogenous nutrient uptake and metabolism [24,31,32,33], but they also activate mammalian target of rapamycin (mTOR)-mediated transcriptional activation of SREBP (sterol regulatory element-binding proteins) for de novo lipid biosynthesis during effector T cell proliferation [34]. Moreover, in order to control T cell overactivation, TCR ligation can activate adenosine 5′-monophosphate-activated protein kinase (AMPK) in Tregs [35], which in turn inhibits T cell clonal overexpansion by enhancing mitochondrial fatty acid oxidation (FAO), reducing mTOR-mediated glycolysis and limiting the production of reactive oxygen species (ROS) via promoting mitophagy [36,37,38,39,40]. Therefore, through regulation of the activity of mTOR and AMPK, T cell extrinsic signals finely coordinate the Yin-Yang balance of intrinsic metabolic pathways (Figure 2), thus orchestrating T cell energy requirements during T cell activation and differentiation.

## 4. Metabolic Programs Determine T Cell Differentiation and Survival

It is worth noting that majority of effector T cells (90%–95%) quickly undergo apoptosis once pathogens are cleared [41]. In contrast to these short-lived effector T cells, human naïve T cells, Tregs and memory T cells have a lifespan of more than 9–10 years [42,43,44]. To dissect the underlying metabolic mechanisms determining T cell fate, it has been shown that activation of mTORC1 promotes the differentiation of Th1, Th17 and follicular T cells, while mTORC2 signaling drives Th2 differentiation [45]. Moreover, activation of the AMPK pathway is critical for the generation of Tregs [46]. As mTOR activation promotes the aerobic glycolysis pathway, whereas AMPK activation favors fatty acid-fueled mitochondrial OXPHOS [47,48], there is an emerging concept that T cell metabolic programs dictate the differentiation and survival of T cells. More specifically, glycolytic metabolism controls the metabolic requirements for short-lived proliferative T cells, such as Th1, Th2, Th17 and CTLs, while mitochondrial OXPHOS is the main metabolic program for long-lived T cell subsets, including naïve T cells, Tregs and memory T cells [30]. In line with this concept, an increase in glycolytic metabolism reduces the lifespan of *C.elegans* [49], whereas FAO extends the lifespan of *D. melanogaster* [50]. Given that T cell-based immunotherapies benefit only a small percentage of cancer patients [51], it remains of intense interest to understand whether modulation of T cell metabolic programs can broaden the clinical application of immunotherapy by improving T cell longevity and anti-tumor function in the TME.

## 5. Modulation of T Metabolic Programs in the TME

Although elevated numbers of CD8^+^ T cells in the TME are positively associated with good prognoses in cancer [52], tumor-killing activity by these CD8^+^ T cells is generally compromised due to the hostile intratumoral metabolic environment in tumors [53]. Aerobic glycolysis is required for IFNγ production in anti-tumor effector CD8^+^ T cells, but rapid consumption by tumor cells restricts the availability of glucose to T cells that dampens glycolysis and IFNγ production, leading to CD8^+^ T cell anergy or exhaustion. Moreover, activated T cells can upregulate the expression of immune checkpoints, including PD-1 and CTLA4, which bind to PD-L1/L2 and CD80/CD86, respectively, on APCs or peripheral tissues to inhibit overactivation and maintain peripheral tolerance [54,55]. In the TME, stromal and tumor cells can harness this mechanism by upregulation of PD-L1/L2 to inhibit anti-tumor responses of effector T cells [56,57]. Recent studies demonstrate that PD-1 signaling switches glycolysis to FAO programs by abrogating the transport and metabolism of glucose and amino acids and enhancing endogenous lipid hydrolysis and mitochondrial transportation. Interestingly, CTLA-4 signaling only inhibits glucose and amino acid metabolism without augmenting FAO [58]. Metabolic rewiring of T cells in the TME provides a cancer immunotherapeutic approach with ICB using anti-PD1/PD-L1 or anti-CTLA-4 antibodies. These antibodies reverse PD-1/CTLA4-mediated glycolytic inhibition, thus reinvigorating the anti-tumor activity of tumor infiltrating CD8^+^ T cells. Despite the benefits of ICB in multiple difficult-to-treat cancer types, the majority of cancers showed primary and acquired resistance to ICB therapy [59,60], suggesting more complicated molecular and cellular mechanisms beyond CD8^+^ T cells mediating tumor immune escape in the TME.

Indeed, tumor microenvironmental signals, such as tumor antigens, cytokines (e.g., TGF-β) and metabolic factors (e.g., lactate and lipid metabolites), can drive naïve CD4^+^ T cell differentiation into Tregs, which are immunosuppressive phenotypes [61]. Tregs are able to suppress anti-tumor immunity through multiple mechanisms, including secreting inhibitory cytokines and interfering with effector CD8^+^ T cell metabolism, differentiation and survival [62]. Recently, Tregs in the TME have been shown to induce effector T cell senescence by upregulating lipid metabolic enzyme group IVA phospholipase A2 (cPLA_2_α). Inhibition of cPLA_2_α activity restores unbalanced T lipid metabolism, thus enhancing their antitumor function [63]. Unlike other effector T cells (e.g., CD8^+^ T cells) in the TME, Tregs mainly rely on FAO to maintain their survival and suppressive function. Thus, inhibition of mitochondrial metabolism and FAO inhibits Treg differentiation and functionality, thus reducing tumor immune evasion in the TME [64,65]. Notably, Tregs do not depend on de novo fatty acid synthesis to fuel FAO. Instead, exogenous fatty acids provide the main lipid source for Treg metabolism [66]. In the TME, exogenous fatty acids also impair antitumor functions of CD8^+^ T cells through inducing lipid peroxidation and ferroptosis [67,68]. Thus, understanding how exogenous fatty acids are taken up, transported and metabolized inside T cells is a new area to determine T cell lipid metabolism, differentiation and function in the TME.

## 6. Fatty Acid Binding Proteins in T Cells

As the backbone of lipids, fatty acids are essential nutrients by constituting membrane structure, providing an energy supply, and functioning as signaling molecules during the lifespan of T cells [69,70]. Fatty acids are divided based on hydrocarbon chain length into very long-chain fatty acids (VLCFAs, 22 or more carbons), long-chain fatty acids (LCFAs, 13–21 carbons), medium-chain fatty acids (MCFAs, 7–12 carbons) and short-chain fatty acids (SCFAs, 1–6 carbons) [71]. Due to their hydrophobic carbon chain, unbound FAs are insoluble in the aqueous environment. As such, a family of fatty acid binding proteins (FABPs) has been evolved to function as fatty acid chaperones, facilitating their uptake, transport and response [72,73,74]. The nine FABP members that have been discovered thus far exhibit tissue-specific distribution patterns. For example, adipose tissue mainly expresses adipose FABP (A-FABP, also known as FABP4), and skin tissue mainly expresses epidermal FABP (E-FABP, or FABP5) [69,75,76]. Accumulating studies from our group and others have demonstrated that FABPs play a central role in regulating cell metabolism and function in multiple disease settings [69,77,78,79,80,81,82].

Based on a microarray analysis of the FABP expression pattern in human leukocyte subsets, macrophages/dendritic cells (DC) mainly express FABP4 and FABP5, whereas T cells predominantly express FABP5 [83]. Single-cell transcriptomic profiling in different T cell subsets confirms that FABP5 is the main FABP isoform expressed across different T cell subsets [84]. Notably, exhausted CD8^+^ T cells seem to have the highest levels of FABP5 (Figure 3A), suggesting that FABP5 might play an important role in regulating the survival and anti-tumor function of CD8^+^ T cells in the TME. FABP5 and FABP4 are highly expressed in tissue resident memory (Trm) T cells in skin, lung and gastric tumors [85,86]. High expression of FABPs increases the uptake of exogenous fatty acids and FAO, which is associated with their prolonged survival and enhanced antitumor immunity. FABP5 is highly expressed in Tregs to maintain their mitochondrial integrity, lipid metabolism and suppressive functions [16,87]. Moreover, our recent studies demonstrate that FABP5 is mainly expressed in T cells, especially naïve CD4^+^ and CD8^+^ T cells (Figure 3B). Expression of FABP5 promotes exogenous linoleic acid uptake and ROS-mediated T cell death [69]. Thus, mounting evidence indicates that FABPs, particularly FABP5, play a previously underappreciated role in regulating T cell lipid metabolism and function in the TME.

## 7. Regulation of T Cell Lipid Metabolism and Function by FABP5

Lipids in T cells mainly come from two sources, endogenous lipogenesis and exogenous uptake from environment/diets. During T cell activation, enhanced uptake of glucose and amino acids can be converted to acetyl-CoA, which is further catalyzed for de novo fatty acid synthesis by acetyl-CoA carboxylase (ACC) and fatty acid synthase (FAS) enzymes to fuel cell proliferation and lineage differentiation [88]. However, in T cell subsets that are less dependent on aerobic glycolysis, such as naïve T cells, Tregs and Trms, the uptake of exogenous fatty acids becomes the main lipid source to fuel their metabolism and function.

Unlike SCFAs or MCFAs, dietary LCFAs cannot easily diffuse through plasma and organelle membranes to be oxidized. As such, FABPs facilitate these processes. It is well known that the mitochondrial carnitine shuttle system, which includes carnitine-palmitoyl transferase 1 (Cpt1), carnitine-acylcarnitine translocase (CACT) and Cpt2, helps the transfer of LCFAs from the mitochondrial outer membrane to the inner membrane for FAO [89]. Cpt1 is the main isoform in T cells, which acts as the rate-limiting step in long-chain FAO [85]. Using FABP*5^−^*^/*−*^ mice, we have shown that the endogenous lipid content in FABP5^−/−^ T cells is not altered, but exogenous fatty acid uptake is reduced in FABP5^−/−^ T cells as compared to FABP5^+/+^ T cells. Moreover, the expression of *Cpt1* is dramatically reduced in *Fabp5*^−/−^ T cells [14]. These data suggest that FABP5 plays a critical role in exogenous LCFA uptake, mitochondrial lipid transport and FAO in T cells.

### 7.1. FABP5 in Naïve T Cells

As long-lived cells, naïve T cells are metabolically quiescent [90]. They utilize the catabolic glucose/mitochondrial OXPHOS pathway to generate ATP for their survival. Interestingly, we recently found that T cell subsets, including naïve T cells, mainly express fatty acid chaperone FABP5 (Figure 3), suggesting that exogenous fatty acids might be involved in naïve T cell metabolism and survival. Given the growing worldwide obesity epidemic, and that obese people have elevated levels of free fatty acids [70,91,92,93], we exposed naïve T cells to various exogenous fatty acid environments and found that exogenous fatty acids can be taken up by naïve T cells to induce mitochondrial oxygen consumption [14]. Interestingly, polyunsaturated fatty acids (such as 18:2 linoleic acid, LA), but not monounsaturated fatty acids (e.g., 18:1 oleic acid, OA), activate mitochondrial ROS production leading to significant naïve T cell death. Consistent with our observations, LA, which is enriched in nonalcoholic fatty liver disease, induces intrahepatic CD4^+^ T cell death via mitochondrial ROS production [94]. In the TME, the uptake of fatty acids by CD36 results in ferroptosis of tumor-infiltrating CD8^+^ T cells [68]. Importantly, using obese mice fed a HFD rich in 18:2 LA, we demonstrate that dietary LA is taken up by naïve T cells and oxidized to generate mitochondrial ROS in a FABP5-dependent manner [14]. As such, FABP5 deficiency successfully rescues LA-indued CD8^+^ T cell death, thus inhibiting mammary tumor growth in the HFD-induce obese mice. These observations suggest a pivotal role for FABP5 in mediating exogenous fatty acid uptake, mitochondrial transport and ROS-induced cell death in naïve T cells.

### 7.2. FABP5 in Tregs

Activation of the AMPK/FAO metabolic pathway is critical for the generation of Tregs [46]. AMPK activation by a high AMP/ATP ratio, AICAR (5-aminoimidazole-4-carboxamide ribonucleotide) or metformin, strongly enhances mitochondrial FAO, thereby promoting the expansion of Tregs [65,95]. In peripheral induced Tregs (iTregs), elevated AMPK activity is likely to modulate Cpt1a activity and increase fatty acid transportation into mitochondria for β-oxidation [96]. The increased expression of the fatty acid translocase CD36 on the surface of Tregs enhances fatty acid uptake, thus facilitating FAO in Tregs [97]. Blocking the transportation of LCFAs into mitochondria by etomoxir (an inhibitor of Cpt1) inhibits Foxp3 expression in Tregs, while exogeneous administration of oleate/palmitate acid promotes Treg generation in vitro [65]. Given the role of FABP5 in exogenous FA uptake and mitochondrial transportation, it is likely that FABP5 plays an important role in maintaining lipid metabolism and function in Tregs.

Indeed, compared to naïve T cells, Tregs exhibit upregulated expression of FABP5. In vitro inhibition of FABP5 decreases mitochondrial cardiolipin synthesis, loss of mitochondrial cristae structure, and FAO, verifying the important role of FABP5 in mitochondrial lipid metabolism and integrity in Tregs. Interestingly, FABP5 inhibition promotes Treg immune suppression by inducing mitochondrial DNA release/type I IFN/IL-10 signaling axis [16]. Consistent with these in vitro observations, Tregs in the TME exhibited an enhanced immunosuppressive activity versus splenic Tregs due to the lack of lipid availability [16]. Our studies using FABP5^−/−^ mice confirm that FABP5 deficiency protects mice from the development of experimental autoimmune encephalomyelitis (EAE) by favoring Treg differentiation and function [98]. Moreover, FABP5^−/−^ mice are associated with an immunosuppressive TME and elevated tumor growth as compared to WT mice [82]. These data collectively support that FABP5 deficiency enhances the immune suppressive function of Tregs, thus favoring tumor evasion and growth.

### 7.3. FABP5 in Memory T Cells

Unlike naïve and effector T cells, resident memory T cells, such as CD69^+^CD103^+^ Trm cells, do not circulate throughout the body [99]. They reside in specific tissues and provide efficient immune responses upon antigen re-exposure [100]. It has been shown that memory T cells rely on FAO rather than glycolysis for their long-term survival [101]. The presence of Trm cells is associated with better outcomes for individuals with ovarian, lung and breast cancers [102,103,104]. However, how FAO enhances Trm survival and anti-tumor immune response remains elusive. A recent study reports that FABPs (mainly FABP4/FABP5), which are highly expressed in skin CD8^+^ Trm cells, mediate exogenous palmitate uptake and mitochondrial FAO in vitro. Deletion of Fabp4/Fabp5 in CD8^+^ Trm cells impairs exogenous free fatty acid uptake, reduces their long-term survival and protective immune responses, suggesting that FABPs play a critical role in the survival and function of CD8^+^ Trm cells [85]. In line with these studies, CD69^+^CD103^+^ Trm cells in the TME of gastric adenocarcinoma highly express PD-1, and PD-1/PD-L1 blockade enhances anti-tumor function of Trm cells by increasing Fabp4/5-mediated lipid uptake and cell survival both in vitro and in vivo [86]. Moreover, CD8^+^ memory T cells rely on lysosomal acid lipase to mobilize FA to fuel mitochondrial FAO [105], and FABP5 has been identified as a key immunometabolic marker in tumor-infiltrating CD8^+^ T cells by promoting FAO and cell survival in human hepatocellular carcinoma [15]. Accumulating evidence reveals a pivotal role for FABP5 in enhancing the longevity and anti-tumor function of memory T cells by facilitating fatty acid uptake and FAO in the TME.

### 7.4. FABP5 in Other T Cell Subsets

As discussed above, naïve T cells undergo dramatic metabolic alterations to develop into distinct effector T cells, including Th1, Th2 and Th17 lineages, among which the mTOR/glycolytic pathway is lineage-decisive, as inhibition of mTOR activity diminishes effector T cell development [106,107]. To accommodate the rapid proliferation of effector T cells, the synthesis of biomolecules, including de novo fatty acid synthesis, appears critical for Th1, Th2 and Th17 development. As ACCs are key enzymes that catalyze the conversion of acetyl-CoA to malonyl-CoA for endogenous synthesis of fatty acids [108], deletion of ACC1 in T cells mainly attenuates the expansion of Th17 and survival of CD8^+^ cells, suggesting that endogenous fatty acid synthesis is important for Th17/CD8^+^ T cells [66,109]. Interestingly, HFD-induced obese mice specifically augment the development of Th17 cells, but not other T cell subsets, through the ACC1/fatty acid synthesis pathway [110]. As FABP5 is associated with lipid elongation and desaturation during the process of endogenous lipid synthesis [16], it is likely that FABP5 expression in T cells favors Th17 cell development via regulating the endogenous lipid synthesis pathway. Indeed, in a mouse EAE model, deficiency of FABP5 protects mice from developing EAE symptoms by reducing Th17 cell differentiation [98]. In contrast, FABP5 expression induces Th17 polarization in both mouse models and human samples in atopic dermatitis [111]. In a *Listeria monocytogenes* infection model, FABP5 deficiency has no impact on the generation/maintenance of antigen-specific effector CD4^+^ and CD8^+^ T cells [112]. Collectively, these studies suggest that although ACC1 expression is generally required for all effector T cell proliferation, compared to Th1 and Th2 cells, Th17 cells appears to rely more on the de novo fatty acid synthesis pathway, which is evidenced by the reduced expansion of Th17 cells rather than other effector subsets (e.g., Th1, Th2 or CTL) in FABP5^−/−^ mice. Given the paradox role of Th17 cells in the TME [113,114], the role of FABP5 in regulating the IL-17 axis in tumor immunotherapy warrants further investigation.

## 8. Conclusions and Future Perspectives

A better understanding of T cell metabolic requirements should improve the anti-tumor responses of T cell-based tumor immunotherapy. There is a growing appreciation that while aerobic glycolysis is essential for effector T cell activation, expansion and function, quiescent T cells (e.g., naïve T cells or memory T cells) are more reliant on mitochondrial OXPHOS for long-term survival. Apart from glucose and amino acids, fatty acids are required for the proper function of Tregs, memory T cells (such as Trm) and Th17 cells. As such, FABP5 has been shown as a new lipid sensor in facilitating fatty acid uptake, lipid signals and mitochondrial FAO in these cells (Figure 4). Accumulating evidence indicates that FABP5 supports survival/expansion not only by promoting exogenous fatty acid uptake, but also by enhancing de novo lipid synthesis and membrane lipid hydrolysis processes [115]. In the TME, FABP5 may be dispensable for effector T cells, which mainly rely on glycolytic pathways, but it is critical in mediating the survival of fatty acid-fueled T cell subsets. Of note, PD-1/PD-L1 ligation inhibits glycolysis but increases fatty acid metabolism in effector T cells [116]. Although inhibiting anti-tumor function, PD-1-mediated metabolic reprogramming enhances FABP5 expression and exhausted T cell survival [58,86], which provides a time window for effective responses of TILs during the treatment of ICB [117]. Thus, FABP5 could represent a novel marker/mechanism for evaluating the effectiveness of ICB treatment in cancer patients.

It is worth noting that obesity is associated with an increased risk of at least 13 types of cancer [118,119]. While FABP4 has been demonstrated to link obesity-associated cancer risk by enhancing the pro-tumor functions of macrophages and adipocytes [69,77,78,120], the role of FABP5 in regulating T cell lipid metabolism and function in obesity-associated cancer risk should also be carefully interrogated. Indeed, FABP5 mediating LA-induced naïve T cell death in obese mouse models supports additional molecular and cellular mechanisms of other FABP family members aside from FABP4 in linking obesity/cancer associations via regulating T cell lipid metabolism and function [14,121]. In addition, while FABP5 expression plays a critical role in maintaining mitochondrial integrity and FAO in Tregs, FABP5 inhibition enhances Treg immunosuppressive function in the TME. More studies are needed for these seemingly paradoxical observations. Besides direct action of FABP5 in T cells, FABP5 is also expressed in macrophages and other APCs in the TME, which indirectly affects T cell infiltration and function [69,82]. In summary, this review highlights an emerging role for FABP5 in regulating T cell lipid metabolism and function in the TME. While FABP5 expression is generally beneficial by promoting FAO-fueled T cell survival in lean patients, obesity-associated lipid dysregulation can promote FABP5-dependent T cell death, leading to impaired anti-tumor responses in obese patients. Understanding of how FABP5 regulates T cell lipid metabolism, survival and function may improve the efficacy of T cell-based immunotherapy to combat cancer in lean and obese patients.

## Figures and Tables

**Figure 1 cancers-15-00657-f001:**
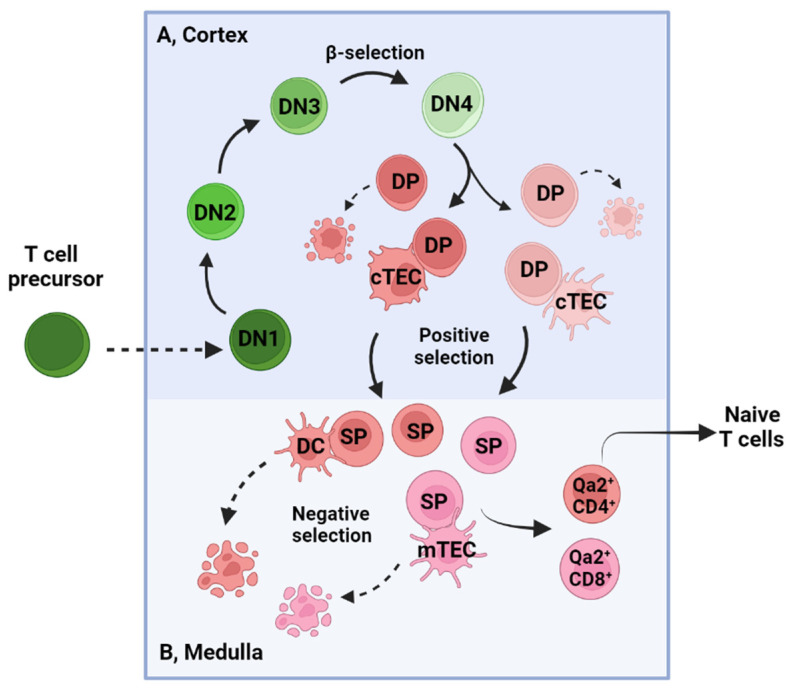
T cell development in the thymus. (**A**) T cell development in the thymic cortex. DN cells (green) undergo four stages from DN1 to DN4 based on CD44 and CD25 expression in the thymic cortex. DN1 cells (CD44^+^CD25^−^) have multiple lineage potencies, but expression of the transcription factor Bcl11b in DN2 cells (CD44^+^CD25^+^) commits them into T-cell lineages (αβ and γδ T cells). The E2A/Notch1 signal transcriptionally controls the pre-T cell receptor (TCR) α expression, and the Myb/Gata3 signal is critical for TCRβ expression. Pre-TCR complex at the DN3 stage (CD44^−^CD25^+^) is formed due to the rearrangement of functional TCRβ and pre-TCR α chains, which is critical for the further expansion and development of αβ T cells. This event is known as β-selection. After passing through β-selection, DN4 (CD44^−^CD25^−^) cells begin to express the TCRα gene, CD4 and CD8, and step into the CD4^+^CD8^+^ DP (red) stage. As the major population in the thymus, DP thymocytes complete TCRα gene rearrangement and form functional TCRαβ, which can recognize major histocompatibility complex (MHC)/self-peptides on the cortical thymic epithelial cells (cTECs). Only those receiving TCR/peptide binding signals can survive and differentiate into SP thymocytes, which is known as positive selection. (**B**) Positive-selected SP thymocytes (less than 5% of DP cells) migrate to the thymic medulla, where cells recognize the MHC/peptide complex on dendritic cells (DCs) or medullary thymic epithelial cells (mTECs). Negative selection occurs when SP thymocytes undergo apoptosis upon receiving high-affinity and long-duration TCR/MHC signals in the medulla. This process is very beneficial for the deletion of self-reactive T cells. After negative selection, SP thymocytes gradually develop into mature Qa2^+^ CD4^+^ or Qa2^+^ CD8^+^ SP stage, which express emigration receptors, such as S1P1, CCR7 and CCR2, and become ready to egress to the peripheral T-cell repertoire.

**Figure 2 cancers-15-00657-f002:**
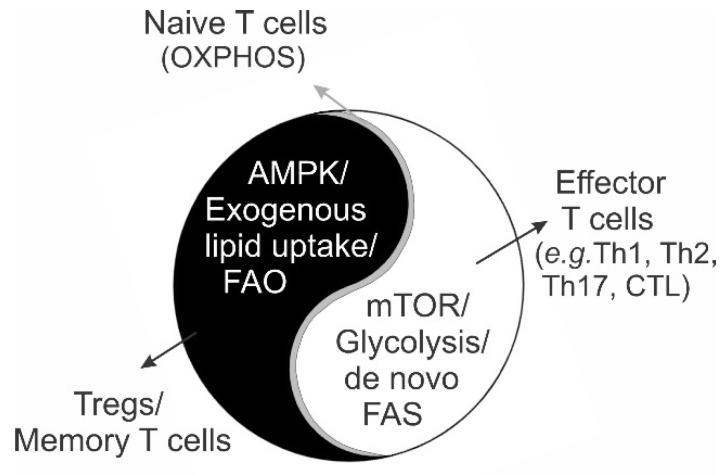
The Yin-Yang balance of T cell metabolism. Naïve T cells remain quiescent by fully metabolizing the basal amount of glucose into ATP, CO_2_ and H_2_O through mitochondrial oxidative phosphorylation (OXPHOS). Once exposed to antigen stimulation (e.g., TCR/CD28 signals), naïve T cells rapidly upregulate the uptake of environmental nutrients (e.g., glucose) via activation of the PI3K/AKT/mTOR pathway and switch to aerobic glycolysis to meet the bioenergetic and biosynthesis needs for the clonal expansion and differentiation of antigen-specific effector T cells (Yang side). To prevent T cell overactivation, Tregs are activated by the AMPK/FAO pathway to suppress effector T cell proliferation during the process (Yin side). Upon antigen clearance, activated T cells can develop into memory T cells, which adopt similar FAO metabolism to maintain their long-term survival. The Yin-Yang balance model of T cell metabolism coined here not only describes the functional and metabolic complements of individual T cell subsets, but also help to explain the overall homeostasis and harmony of T cell responses in vivo as a whole.

**Figure 3 cancers-15-00657-f003:**
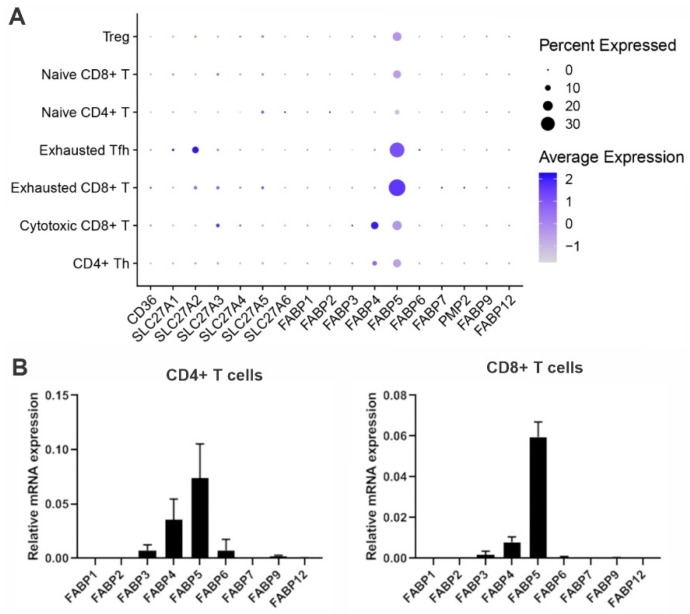
Expression profile of FABP family members in T cells. (**A**) Analysis of the expression profile of fatty acid uptake related genes, including CD36, fatty acid transport protein 1-6 (FATP 1-6, encoded by SLC27A1-6) and FABP family members, in different T cell subsets using a publicly accessible dataset GSE131907. (**B**) Expression of FABP family members in naïve CD4^+^ and CD8^+^ T cells. Naïve CD4^+^ and CD8^+^ T cells were purified from the spleen of C57 BL/6 mice using a flow sorter. RNA was extracted for real-time PCR analysis of FABP family members.

**Figure 4 cancers-15-00657-f004:**
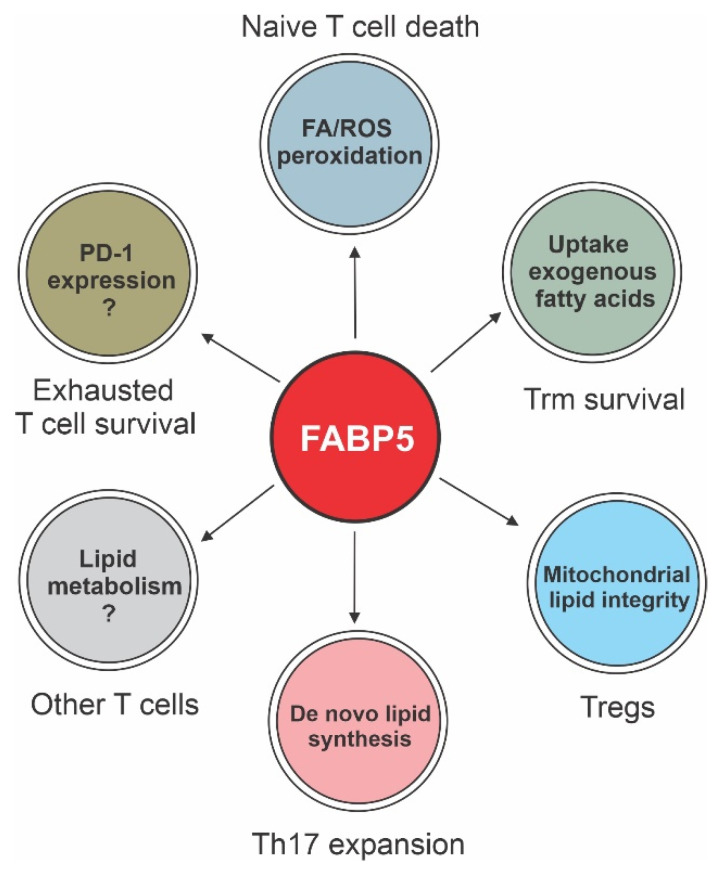
Role of FABP5 in regulating lipid metabolism in different subsets of T cells. As the main lipid chaperone in T cells, FABP5 facilitates the uptake of exogenous fatty acids and *de novo* lipid synthesis, thus promoting mitochondrial FAO and membrane lipid synthesis, respectively. In general, FABP5 deficiency is associated with enhanced immune suppressive activity of Tregs and reduced survival of Trm and exhausted T cells in the TME, which contribute to elevated tumor growth in FABP5^-/-^ mice on the LFD. Notably, naïve T cells mainly rely on glucose for their survival, while uptake of exogenous fatty acids, especially LA, by FABP5 promotes mitochondrial ROS and lipid peroxidation, inducing naïve T cell death; thus, mice on HFD rich in LA exhibit reduced CD8^+^ T cells in the TME, leading to excecated tumor growth. The role of FABP5 in mediating lipid metabolism and function in other T cell subsets (e.g., Th17 or γδ T cells) in the TME needs further investigation.

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
