# Peer review of "Role of FABP5 in T Cell Lipid Metabolism and Function in the Tumor Microenvironment"

_cancers, 2023, doi:10.3390/cancers15030657_

Round 1

Reviewer 1 Report

Rong Jin et.al did a great job describing the metabolic rewiring during T-cell activation and PD-1-related signaling modulation and highlighted the potential of FABP5 and its associated lipid metabolism in the tumor microenvironment modulation. This review is generally well-written. I would suggest publishing after addressing the following minor concerns.

  1. The title seems not representative for this review paper, sounds more like a title for a research paper. This review generally discuss the metabolic rewiring in the tumor microenviroment with a focus of the potential of FABP5's role, not directly show the effects of FABP5 in regulating tumor microenviroment. It is suggested to make some slight modification to represent the content of this review paper.
  2. Although the title highlighted lipid metabolism, the role of regulating lipid/fatty acid metabolism in treating tumors is not strenthened enough. Despite the authors did discussed about this in the section 5: Modulation of T Metabolic Programs in the TME, it feels not strong enough. Consider putting more efforts through adding extra published works in this part.
  3. It is suggested to include the dataset source of the scRNAseq analysis shown in Figure 3 by either citing their own paper or public datasets.

Author Response

Rong Jin et.al did a great job describing the metabolic rewiring during T-cell activation and PD-1-related signaling modulation and highlighted the potential of FABP5 and its associated lipid metabolism in the tumor microenvironment modulation. This review is generally well-written. I would suggest publishing after addressing the following minor concerns.

Response: We greatly appreciated the positive comments!

1. The title seems not representative for this review paper, sounds more like a title for a research paper. This review generally discuss the metabolic rewiring in the tumor microenviroment with a focus of the potential of FABP5's role, not directly show the effects of FABP5 in regulating tumor microenviroment. It is suggested to make some slight modification to represent the content of this review paper.

Response: Thanks for the suggestion. We modified the title as suggested.

2. Although the title highlighted lipid metabolism, the role of regulating lipid/fatty acid metabolism in treating tumors is not strenthened enough. Despite the authors did discussed about this in the section 5: Modulation of T Metabolic Programs in the TME, it feels not strong enough. Consider putting more efforts through adding extra published works in this part.

Response: We have added more works in this part as suggested.

3. It is suggested to include the dataset source of the scRNAseq analysis shown in Figure 3 by either citing their own paper or public datasets.

Response: We have included the source reference for the scRNA seq in Figure 3.

Thanks!

Reviewer 2 Report

The authors did an excellent job assembling this review. It was very pleasant reading through this well drafted manuscript. Deftly covered T-cell development, activation, and metabolic regulation of this process. Aptly introduces the role of Fatty acid metabolism and fatty acid binding proteins in T-cells. Overall, they very clearly discuss about regulation of T-Cell Lipid Metabolism and role of FABP5. I commend the authors for this excellent review.

Minor correction

Line 178

Remove extra Y after “…immunotherapeutic approach y with…”

Author Response

The authors did an excellent job assembling this review. It was very pleasant reading through this well drafted manuscript. Deftly covered T-cell development, activation, and metabolic regulation of this process. Aptly introduces the role of Fatty acid metabolism and fatty acid binding proteins in T-cells. Overall, they very clearly discuss about regulation of T-Cell Lipid Metabolism and role of FABP5. I commend the authors for this excellent review.

Minor correction

Line 178

Remove extra Y after “…immunotherapeutic approach y with…”

Response: Thanks so much for the positive comments of the Reviewer 2. We have removed the “y” as pointed out.

Thanks!

Reviewer 3 Report

This manuscript is a review of how T cell lipid metabolism affects activation and function in the tumor microenvironment with a focus on the role of FABP5. This is a timely article since immunometabolism is of recent interest, particularly for its application in cancer therapy. This article describes T cell metabolic changes throughout development and how different metabolic pathways are critical for T cell function. The focus on FABP5 in the T cell microenvironment is novel and has not been previously reviewed in this level of detail. This manuscript should be accepted with the below minor changes.

Minor points

1. The authors should clarify why the change from green (DN) to red (DP) T cells in figure 1 legend as it is currently unspecified.

2. Page 3, line 102 and line 104. The authors note that external signals lead to T cell activation via metabolic pathways. This should be stated with more nuance since T cell activation also occurs through additional mechanisms as the authors explain later in the paragraph on Page 5, line 185. 

3. There is recent data concerning the role of lipid metabolism in T cell senescence (PMID33790024 and others). Brief discussion of these findings should be added to section 4 on Page 4. 

4. There is significant data concerning the role of FABP5 in APC function as noted by the authors. The authors should comment of how altered APC function by FABP5 may affect T cell activation in addition to the direct action of FABP5 in T cells. 

Author Response

This manuscript is a review of how T cell lipid metabolism affects activation and function in the tumor microenvironment with a focus on the role of FABP5. This is a timely article since immunometabolism is of recent interest, particularly for its application in cancer therapy. This article describes T cell metabolic changes throughout development and how different metabolic pathways are critical for T cell function. The focus on FABP5 in the T cell microenvironment is novel and has not been previously reviewed in this level of detail. This manuscript should be accepted with the below minor changes.

Minor points

  1. The authors should clarify why the change from green (DN) to red (DP) T cells in figure 1 legend as it is currently unspecified.

Response: As there are multiple T cell differentiation stages in the cortex. Color change in green and red for individual stage is easier for readers. We have specified it in the legend as suggested. 

  1. Page 3, line 102 and line 104. The authors note that external signals lead to T cell activation via metabolic pathways. This should be stated with more nuance since T cell activation also occurs through additional mechanisms as the authors explain later in the paragraph on Page 5, line 185. 

Response: We have reworded the description to make it more accurate. Thanks.

  1. There is recent data concerning the role of lipid metabolism in T cell senescence (PMID: 33790024 and others). Brief discussion of these findings should be added to section 4 on Page 4. 

Response: I have included this seminal study and discussed it as suggested.

  1. There is significant data concerning the role of FABP5 in APC function as noted by the authors. The authors should comment of how altered APC function by FABP5 may affect T cell activation in addition to the direct action of FABP5 in T cells. 

Response: Yes, we have included this comment at the end of the review.

Thanks so much!